Physiological response of the cold-water coral Desmophyllum dianthus to thermal stress and ocean acidification

Gori Andrea 1 2 agori.mail@gmail.com
Ferrier-Pagès Christine 2
Hennige Sebastian J. 1
Murray Fiona 1
Rottier Cécile 2
Wicks Laura C. 1
Roberts J. Murray 1
1 Centre for Marine Biodiversity and Biotechnology, Heriot-Watt University , Edinburgh, Scotland , United Kingdom
2 Coral Ecophysiology, Centre Scientifique de Monaco , Monaco , Principality of Monaco
Medina Mónica
Electronic publication date: 2016 Feb 2
Publication date: 2016
Volume: 4
Electronic Location ID: e1606
Received 2015 Jul 28; Accepted 2015 Dec 27
Copyright: ©2016 Gori et al.
Copyright year: 2016
Copyright holder: Gori et al.
License: This is an open access article distributed under the terms of the Creative Commons Attribution License, which permits unrestricted use, distribution, reproduction and adaptation in any medium and for any purpose provided that it is properly attributed. For attribution, the original author(s), title, publication source (PeerJ) and either DOI or URL of the article must be cited.
License URL: https://creativecommons.org/licenses/by/4.0/

Keywords: Cold-water corals, Thermal stress, Ocean acidification, Coral calcification, Coral respiration, Coral excretion

Funding: UK Natural Environment Research Council NE/J021121/1 NE/H017305/1 NE/K009028/1 Government of the Principality of Monaco Heriot-Watt University’s Environment and Climate Change theme Marine Alliance for Science and Technology Scotland (MASTS) This work was supported by the UK Natural Environment Research Council (grants NE/J021121/1 and NE/H017305/1 to JMR, NE/K009028/1 to SJH) and the Government of the Principality of Monaco. JMR, LCW and SJH received additional support from Heriot-Watt University’s Environment and Climate Change theme and the Marine Alliance for Science and Technology Scotland (MASTS). The funders had no role in study design, data collection and analysis, decision to publish, or preparation of the manuscript.

==============================
Rising temperatures and ocean acidification driven by anthropogenic carbon emissions threaten both tropical and temperate corals. However, the synergistic effect of these stressors on coral physiology is still poorly understood, in particular for cold-water corals. This study assessed changes in key physiological parameters (calcification, respiration and ammonium excretion) of the widespread cold-water coral Desmophyllum dianthus maintained for ∼8 months at two temperatures (ambient 12 °C and elevated 15 °C) and two pCO2 conditions (ambient 390 ppm and elevated 750 ppm). At ambient temperatures no change in instantaneous calcification, respiration or ammonium excretion rates was observed at either pCO2 levels. Conversely, elevated temperature (15 °C) significantly reduced calcification rates, and combined elevated temperature and pCO2 significantly reduced respiration rates. Changes in the ratio of respired oxygen to excreted nitrogen (O:N), which provides information on the main sources of energy being metabolized, indicated a shift from mixed use of protein and carbohydrate/lipid as metabolic substrates under control conditions, to less efficient protein-dominated catabolism under both stressors. Overall, this study shows that the physiology of D. dianthus is more sensitive to thermal than pCO2 stress, and that the predicted combination of rising temperatures and ocean acidification in the coming decades may severely impact this cold-water coral species.

Introduction

Increases in anthropogenic carbon emissions, leading to rising sea temperatures and ocean acidification, have resulted in extensive tropical coral bleaching (e.g., Hoegh-Guldberg, 1999; Mcleod et al., 2013) and decreased coral calcification rates (e.g., Gattuso et al., 1998; Chan & Connolly, 2012; Movilla et al., 2012; Bramanti et al., 2013). The combination of rising temperatures and ocean acidification are substantial threats for corals in the next few decades (Hoegh-Guldberg et al., 2007; Silverman et al., 2009; Erez et al., 2011). While considerable research efforts have focused on tropical and temperate corals, less is known about the effects of ocean warming and acidification on cold-water corals (CWC) (e.g., Guinotte et al., 2006; Rodolfo-Metalpa et al., 2015 and references therein). These corals are among the most important ecosystem engineering species (sensu Jones, Lawton & Shachak, 1994) in the deep sea, where they build three-dimensional frameworks (Roberts, Wheeler & Freiwald 2006) that support a highly diverse associated fauna (Henry & Roberts, 2007; Buhl-Mortensen et al., 2010). Scleractininan CWC are most commonly distributed at temperatures between 4 °C and 12 °C (Roberts, Wheeler & Freiwald, 2006), and show species-specific responses to temperatures above their natural thermal range. For instance, elevated seawater temperatures increased calcification in the non-reef forming Dendrophyllia cornigera (Naumann, Orejas & Ferrier-Pagès, 2013; Gori et al., 2014a); had no effect on calcification in the solitary coral Desmophyllum dianthus (Naumann, Orejas & Ferrier-Pagès, 2013); and had either no effect on the reef-forming Lophelia pertusa calcification (Hennige et al., 2015) or induced mortality (Brooke et al., 2013) depending upon the site of origin and change in temperature.

In comparison to thermal stress, CWC seem to have a general capacity to withstand ocean acidification under experimental time periods of up to 12 months. Decreases in pH did not affect calcification rates in both the reef forming L. pertusa and Madrepora oculata (Form & Riebesell, 2012; McCulloch et al., 2012; Maier et al., 2012; Maier et al., 2013a; Hennige et al., 2014; Hennige et al., 2015; Movilla et al., 2014a), or the non-reef forming D. cornigera, D. dianthus (Movilla et al., 2014b; Rodolfo-Metalpa et al., 2015), Caryophyllia smithii (Rodolfo-Metalpa et al., 2015) or Enallopsammia rostrata (McCulloch et al., 2012). However, whether calcification can be sustained indefinitely remains unclear, as seawater acidification has been shown to affect coral metabolism (Hennige et al., 2014), increasing energy demand (McCulloch et al., 2012), and leading to up-regulation of genes related to stress and immune responses, energy production and calcification (Carreiro-Silva et al., 2014). Coral responses to ocean acidification may also depend on seawater temperature (e.g., Reynaud et al., 2003; Edmunds, Brown & Moriarty, 2012), and evidence is now emerging that only when these two factors are combined (as is likely with future climatic changes), do the real effects of ocean change become apparent (Reynaud et al., 2003; Roberts & Cairns, 2014).

This study focused on the combined effects of increased temperature and pCO2 on key physiological processes of the cosmopolitan solitary CWC D. dianthus (Cairns & Zibrowius, 1997) sampled in the deep waters of the Mediterranean Sea. Calcification, respiration, and ammonium excretion were quantified in corals maintained over ∼8 months under a combination of conditions that replicated ambient temperature and pCO2 levels (12 °C—390 ppm, Movilla et al., 2014b), and elevated temperature and pCO2 levels predicted in the IPCC IS92a emission scenarios (15 °C—750 ppm, following Riebesell et al., 2010). We hypothesize that the combination of elevated temperatures and pCO2 will have a greater impact on coral calcification, respiration and excretion than single stressors. Analysis of the ratio of respired oxygen to excreted nitrogen (O:N), which is a physiological index providing information on the main sources of energy being metabolized (Sabourin & Stickle, 1981; Yang et al., 2006; Zonghe et al., 2013), was used to reveal whether corals are mainly metabolizing proteins, carbohydrates or lipids, giving a further indication of coral stress under the experimental conditions.

Figure 1 The cold-water coral Desmophyllum dianthus.

Photo by A Gori.

Materials and Methods

Coral collection and maintenance

Specimens of D. dianthus (Esper, 1794) (Fig. 1) were collected in the Bari Canyon (Adriatic Sea, Mediterranean Sea, 41°17.2622′N, 17°16.6285′E, 430 m depth) by the Achille M4 and Pollux III ROVs, and kept alive on board the RV ‘Urania’ during the cruise ARCADIA (March 2010). Corals were transported to the Centre Scientifique de Monaco (CSM, Monaco, Principality of Monaco, CITES permit 2012MC/7725) and maintained there for ∼35 months in 50 L continuous flow-through tanks, with seawater pumped from 50 m depth at a rate of 20 L h−1. Water temperature was maintained close to in situ conditions (12 ± 1.0 °C), and powerheads provided continuous water movement within the tanks. Corals were fed five times a week with frozen Mysis (Crustacea, Eumalacostraca) and adult Artemia salina (Crustacea, Sarsostraca). For experimental work, 12 specimens of D. dianthus were transferred to Heriot-Watt University (Edinburgh, Scotland, UK, CITES permit 2012MC/7929), and kept under collection site ambient conditions for ∼2 months before beginning the experimental incubations. Corals were then placed into ambient temperature and pCO2 (12 °C—390 ppm) levels, and predicted future conditions following the IPCC IS92a emission scenarios (Riebesell et al., 2010): ambient temperature and elevated pCO2 (12 °C—750 ppm), elevated temperature and ambient pCO2 (15 °C—390 ppm), and elevated temperature and pCO2 (15 °C—750 ppm).

For each treatment, there were three replicate systems of ∼80 L tanks, holding one coral each. The tanks were equipped with pumps and filtration units to ensure adequate water mixing and filtration. Tanks were closed systems, filled with seawater collected from the east coast of Scotland (St. Andrews), with partial water changes (20%) every two weeks. Ambient and mixed elevated pCO2 air mixes were bubbled directly into the tanks as described by Hennige et al. (2015). Gas mixing was achieved to target levels, by mixing pure CO2 with air plumbed from outside of the laboratory building in mixing vessels. Mixed or ambient gas was then supplied to appropriate experimental systems. Target gas levels were checked and adjusted daily using a LI-COR 820 gas analyzer calibrated using pre-mixed 0 and 750 CO2 ppm gases (StG gases). All replicate systems were housed in darkness within a temperature-controlled room at 9 °C ± 0.5 °C, and water temperatures in the systems (12 °C ± 0.5 °C and 15 °C ± 0.5 °C) were controlled through Aqua Medic T-computers and titanium heaters (Aqua Medic TH-100). Experimental system temperature, salinity (YSI 30 SCT) and pH(NBS) (Hach HQ 30D) were measured and recorded throughout the duration of experiment. Average pH(NBS) (±standard deviation) values for each treatment (pooled between 3 replicate tanks) over this 8 month period were: 12°C—380ppm = 7.96 ± 0.06; 12°C—750ppm = 7.92 ± 0.06; 15°C—380ppm = 7.97 ± 0.04; and 15°C—750ppm = 7.90 ± 0.06. Further details about the incubation systems are available in Hennige et al. (2015), which support routine pH(NBS) measurements and highlight the stability of these systems over prolonged time periods (Table S1). Corals were fed 3 times a week with a controlled supply of 2 krill (Gamma frozen blister packs) per polyp per feeding event.

Physiological measurements

After 236 days under experimental conditions, four sets of incubations were performed, one for each experimental condition to assess rates of calcification, respiration and ammonium excretion. Each incubation started with the preparation of 1 L of 50 µm pre-filtered seawater. 140 ml of this seawater was sampled for the initial determination of the total alkalinity (TA) (120 ml) and ammonium concentration (20 ml) as described below. The remaining filtered water was equally distributed between 4 incubation chambers (200 ml each). One chamber was left without a coral polyp and used as a control. Three other chambers housed one polyp, each from a different replicate system. Polyps were incubated for six hours in the individual chambers that were completely filled (without any air space) and hermetically closed, according to the standardized protocol developed by Naumann et al. (2011). Constant water movement inside the beakers was ensured by a teflon-coated magnetic stirrer. At the end of the incubation, 140 ml of seawater was taken from each incubation chamber and split between storage vessels for the determination of the final TA and ammonium concentration as described below.

Coral calcification rates were assessed using the alkalinity anomaly technique (Smith & Key, 1975; Langdon, Gattuso & Andersson, 2010), assuming a consumption of 2 moles of alkalinity for every mole of calcium carbonate produced (Langdon, Gattuso & Andersson, 2010). Seawater samples (120 ml) from before and after incubation, were sterile filtered (0.2 µm) and fixed with HgCl2 to prevent further biological activity. TA was determined on 6 subsamples of 20 ml from each chamber using a titration system composed of a 20 ml open thermostated titration cell, a pH electrode calibrated on the National Bureau of Standards scale, and a computer-driven titrator (Metrohm 888 Titrando, Riverview, FL, USA). Seawater samples were kept at a constant temperature (25.0 ± 0.2 °C) and weighed (Mettler AT 261, L’Hospitalet de Llobregat, Spain, precision 0.1 mg) before titration to determine their exact volume from temperature and salinity. TA was calculated from the Gran function applied to pH variations from 4.2 to 3.0 as the function of added volume of HCl (0.1 mol L−1), and corrected for changes in ammonium concentration resulting from metabolic waste products (Jacques & Pilson, 1980; Naumann et al., 2011). Change in the TA measured from the control chamber was subtracted from the change in TA in the chambers with corals, and calcification rates were derived from the depletion of TA over the 6 h incubation.

Respiration rates were assessed by measuring oxygen concentration in the incubation chambers during incubations with optodes (OXY-4 micro, PreSens, Germany) calibrated using sodium sulfite and air saturated water as 0 and 100% oxygen saturation values, respectively. Variations in oxygen concentrations measured from the control chamber were subtracted from those measured in the coral chambers, and respiration rates were derived from the recorded depletion of dissolved oxygen over the incubation. Oxygen consumption rates were converted to C equivalents (µmol) according to the equation C respired =O2consumed ⋅RQ, where RQ is a coral-specific respiratory quotient equal to 0.8 mol C/mol O2 (Muscatine, McCloskey & Marian, 1981; Anthony & Fabricius, 2000; Naumann et al., 2011).

Excretion rates were assessed by determining ammonium concentration in seawater samples (20 ml) that were sterile filtered (0.2 µm) and kept frozen (−20 °C) until ammonium concentration was determined in 4 replicates per sample through spectrofluorometric techniques (Holmes et al., 1999, protocol B).

Results from calcification, respiration and ammonium excretion measurements were normalized to the coral skeletal surface area (fully covered by coral tissue), to allow for comparison with other coral species. The skeletal surface area (S) of each coral polyp was determined by means of Advanced Geometry (Naumann et al., 2009) according to the equation S = π ⋅ (r + R) ⋅ a + π ⋅ R2, where r and R represent the basal and apical radius of each polyp respectively, and a is the apothem measured with a caliper (Rodolfo-Metalpa et al., 2006). Finally, the O:N ratio was calculated for each coral from the results of the measured oxygen respired and ammonium excreted in atomic equivalents (Yang et al., 2006; Zonghe et al., 2013).

Statistical analyses

All results were expressed as means ± standard error. Normal distribution of the residuals was tested using a Shapiro–Wilk test performed with the R-language function shapiro.test of the R 3.1.2 software platform (R Core Team, 2014). Homogeneity of variances was tested by the Bartlett test performed with the R-language function bartlett.test. Differences in the variation of TA, oxygen and ammonium concentration between control and experimental chambers were tested by means of a Wilcoxon–Mann–Whitney test performed with the R-language function wilcoxon.test. Differences among the four experimental conditions in calcification, respiration, ammonium excretion, and O:N ratio were tested by two-way ANOVA with temperature (12 °C–15 °C) and pCO2 (390 ppm–750 ppm) as factors, performed with the R-language function aov.

Table 1 Two-way ANOVA for comparison of calcification, respiration, ammonium excretion rates, and O:N ratio among the experimental treatments; significant p-values are indicated with one (p-value < 0.05), two (p-value < 0.01), or three asterisks (p-value < 0.001).

		F	p value		
Calcification	Temperature	8.58	0.019	*	
	pCO2	1.89	0.206		
	Temperature:pCO2	0.44	0.524		
Respiration	Temperature	1.04	0.337		
	pCO2	0.29	0.602		
	Temperature:pCO2	12.44	0.008	**	
Ammonium excretion	Temperature	1.01	0.344		
	pCO2	0.06	0.811		
	Temperature:pCO2	2.07	0.188		
O:N	Temperature	0.69	0.431		
	pCO2	0.48	0.509		
	Temperature:pCO2	7.94	0.023	*	

Results

TA changes in incubation chambers (2.8–12.8 µEq L−1 h−1) were consistently higher (Wilcoxon–Mann–Whitney test, U = 48, p = 0.004) than changes measured in the control chambers (<0.5 µEq L−1 h−1). Regardless of pCO2 level, calcification rates assessed with the TA anomaly technique (Fig. 2A) were significantly lower in corals maintained at 15 °C compared to those maintained at 12 °C (ANOVA, F = 8.57, p = 0.019, Table 1). For each temperature treatment assessed individually, calcification did not significantly differ at either pCO2 level.

Figure 2 Main physiological processes in Desmophyllum dianthus under the two experimental temperatures (12 and 15 ° C) and the two pCO2 (390 and 750 ppm).

(A) Calcification rate, (B) respiration rate, and (C) ammonium excretion rate as the result of coral nubbins incubation in individual beakers for 6 h. Values are presented as means ± s.e. normalised to coral skeletal surface area.

Oxygen depletion from coral respiration in incubation chambers (5.3–54.7 µmol L−1h−1) was significantly higher (Wilcoxon–Mann–Whitney test, U = 47, p = 0.002) than oxygen depletion in the control chambers from microbial respiration (<4.2 µmol L−1h−1). Respiration rates (Fig. 2B) of corals kept under increased temperature and pCO2 were significantly lower compared to other treatments (ANOVA, F = 12.44, p = 0.007, Table 1).

Changes in ammonium concentration from coral excretion in incubation chambers (0.39–1.78 µmol L−1h−1) were significantly higher (Wilcoxon–Mann–Whitney test, U = 48, p = 0.001) than changes in control chambers from microbial activity (<0.04 µmol L−1h−1). Coral excretion rates (Fig. 2C) were not significantly different among treatments (Table 1).

The ratio of respired oxygen to excreted nitrogen (O:N) (Fig. 3) in corals kept under increased temperature and pCO2 was significantly lower than in the other treatments (ANOVA, F = 7.94, p = 0.023, Table 1).

Figure 3 Ratio of respired oxygen to excreted nitrogen (O:N) of Desmophyllum dianthus under the two experimental temperatures (12 and 15 ° C) and the two pCO2 levels (390 and 750 ppm).

Values are presented as means ± s.e. normalized to coral skeletal surface area.

Discussion

Overall, the results of this study show that the CWC D. dianthus is more sensitive to changes in temperature than to ocean acidification stress. This CWC maintains its metabolism under elevated pCO2, whereas calcification is significantly reduced under elevated temperatures. Furthermore, there is a clear synergistic impact when elevated temperature and pCO2 are combined, resulting in a severe reduction of coral metabolism.

D. dianthus has the ability to withstand elevated pCO2 (750 ppm) under ambient temperature ( °C) over ∼8 months, with no change in calcification, respiration and ammonium excretion rates (Fig. 2 and Table 1). This agrees with previous studies on the same species (Movilla et al., 2014b; Carreiro-Silva et al., 2014; Rodolfo-Metalpa et al., 2015), and with the general consensus that CWC can physiologically cope with elevated pCO2 in the mid-term (3–12 months, Form & Riebesell, 2012; Maier et al., 2013a; Maier et al., 2013b; Movilla et al., 2014a; Hennige et al., 2015). This may be due to their ability to buffer external changes in seawater pH by up-regulating their pH at the site of calcification (McCulloch et al., 2012; Anagnostou et al., 2012), therefore allowing calcification even in aragonite-undersaturated seawater (Venn et al., 2013). Increased expression of genes involved in cellular calcification and energy metabolism may indicate the mechanisms by which D. dianthus continues to calcify under elevated pCO2 at rates similar to those recorded at ambient pCO2 (Carreiro-Silva et al., 2014). Whereas microdensity and porosity of D. dianthus skeleton have been shown to be unaffected by increased pCO2 (Movilla et al., 2014b), the effects of elevated pCO2 conditions on hidden skeleton microstructure and aragonitic crystals organisation cannot be discounted (e.g., molecular bond lengths and orientation, see Hennige et al., 2015). Such effects would take a long time to become evident as reduced skeletal microdensity and porosity, due to the very slow growth rates of D. dianthus (Orejas et al., 2011; Naumann et al., 2011). The experimentally observed physiological ability of D. dianthus to cope with elevated pCO2 is also supported by the recent observation of this CWC in aragonite-undersaturated waters (Thresher et al., 2011; McCulloch et al., 2012; Jantzen et al., 2013a; Fillinger & Richter, 2013). However, there is the possibility that high food availability in these areas may allow corals to sustain the cost of calcification under low pH (Jantzen et al., 2013a; Fillinger & Richter, 2013).

In contrast to elevated pCO2, elevated temperature alone significantly reduced calcification in D. dianthus (Fig. 2B and Table 1). Calcification shows a strong sensitivity to temperature in this CWC species (McCulloch et al., 2012), which is able to maintain growth under elevated seawater temperatures for a short time (3 months at 17.5 °C, Naumann, Orejas & Ferrier-Pagès, 2013), but when exposed to thermal stress for longer periods (∼8 months at 15 °C, this study) calcification rates are significantly reduced. Decreased calcification in D. dianthus under prolonged elevated temperature might be linked to decreased activity in the enzymes involved in calcification (such as carbonic anhydrases; Ip, Lim & Lim, 1991; Al-Horani, Al-Moghrabi & De Beer, 2003; Allemand et al., 2004), since enzyme activity is maximal within the thermal range of the species and decreases otherwise (Jacques, Marshall & Pilson, 1983; Marshall & Clode, 2004; Al-Horani, 2005). Reported calcification rates by D. dianthus have varied widely between studies. Rates measured here (1.26 ± 0.20 µmol CaCO3 cm−2 d−1) were in the same order of magnitude as the rates reported by Naumann et al. (2011) in the Mediterranean (∼3.84 µmol CaCO3 cm−2 d−1), and much lower than those reported by Jantzen et al. (2013b) in Chilean fjords (18.6–54.4 µmol CaCO3 cm−2 d−1). Whilst direct comparison with other studies is problematic due to differences in methodology (total alkalinity vs buoyant weight) or normalization techniques, the rates measured here are consistent with previous results from Mediterranean D. dianthus (e.g., Orejas et al., 2011; Maier et al., 2012; Movilla et al., 2014b), and are much higher than rates measured in D. dianthus from Azores (Carreiro-Silva et al., 2014). Differences in the quality and quantity of food provided to corals (Mortensen, 2001; Jantzen et al., 2013b), coral size (Carreiro-Silva et al., 2014; Movilla et al., 2014b), or intraspecific variability and local adaptation could all contribute to observed variability between studies.

The synergistic effects of elevated temperature and pCO2 on calcification, respiration and O:N ratio observed in this study (Fig. 2 and Table 1), show that these stressors interact to control D. dianthus metabolism causing a far greater effect than increased temperature or pCO2 in isolation (Reynaud et al., 2003). Under elevated temperature and pCO2 treatment, respiration dropped to low values (1.2 ± 0.7 µmol C cm−2 d−1) comparable to those reported for starved D. dianthus (∼1.5 µmol C cm−2d−1, Naumann et al., 2011) or for D. dianthus fed only twice a week (1.34 ± 0.31 µmol C cm−2d−1, Gori et al., 2014b), indicating a reduction in the coral’s metabolic activity. Reduced metabolism is reflected in the concurrent significant reduction in calcification rates (Fig. 2A). Whilst ammonium excretion, which results from protein and amino acid catabolism (Wright, 1995; Talbot & Lawrence, 2002), was not significantly affected by either or both elevated temperature and pCO2 (consistent with previous studies, Carreiro-Silva et al., 2014), the combined effects of elevated temperature and pCO2 caused a shift in O:N from ∼30 to ∼13 (Fig. 3). This highlights a shift from a mixed use of protein and carbohydrate or lipid, to a much less efficient protein-dominated catabolism for energy (Pillai & Diwan, 2002) indicating metabolic stress (Zonghe et al., 2013). Conversely, single stressors caused a slightly increase in O:N ∼30 to ∼50. This is a consequence of increased respiration combined with steady ammonium excretion, leading to a shift to a carbohydrate or lipid-dominated metabolism (Sabourin & Stickle, 1981; Uliano et al., 2010; Zonghe et al., 2013). This is a possible way for the corals to fulfill increased energy demands needed to maintain cell homeostasis under single stressors, but this may be insufficient when subjected to multiple stressors.

Overall, this study shows that the combined effects of increased temperature and pCO2 result in a significant change in D. dianthus metabolism. This may represent an immediate threat to CWC as their habitats are expected to be exposed to both high temperature events and reduced seawater pH with increased frequency in the near future (Roberts & Cairns, 2014). Given the major role of feeding on the metabolism of CWC species (Naumann et al., 2011), it is also extremely important to understand how coral responses to single or multiple stressors can be affected by food availability and quality (Dodds et al., 2007; Thomsen et al., 2013; Rodolfo-Metalpa et al., 2015). Reduced food availability will limit the allocation of extra-energy to physiological adjustments under stress conditions, which could further heighten the negative impacts of elevated temperature and pCO2 on coral metabolism. Studies into the combined impact of climate change and changes in food quantity and quality would provide a more holistic insight into the future of CWC in a changing ocean.

Supplemental Information

Data S1 Raw data

Click here for additional data file.

Table S1 Chemical and physical properties of seawater systems (mean ±1 SD) from October 2014 to July 2015.

Temperature, salinity, pH, alkalinity and DIC (dissolved inorganic carbon) were measured. All other values (pCO2; Ωaragonite, aragonite saturation state; Ωcalcite, calcite saturation state) were calculated using CO2 calc (Robbins et al., 2010) from pH, alkalinity and DIC.

Click here for additional data file.

Additional Information and Declarations

Competing Interests

Author Contributions

Field Study Permissions

Data Availability

The authors declare there are no competing interests.

Andrea Gori conceived and designed the experiments, performed the experiments, analyzed the data, wrote the paper, prepared figures and/or tables, reviewed drafts of the paper.

Christine Ferrier-Pagès conceived and designed the experiments, analyzed the data, contributed reagents/materials/analysis tools, wrote the paper, reviewed drafts of the paper.

Sebastian J. Hennige and Laura C. Wicks wrote the paper, reviewed drafts of the paper.

Fiona Murray analyzed the data, wrote the paper, prepared figures and/or tables, reviewed drafts of the paper.

Cécile Rottier performed the experiments, reviewed drafts of the paper.

J. Murray Roberts conceived and designed the experiments, contributed reagents/materials/analysis tools, wrote the paper, reviewed drafts of the paper.

The following information was supplied relating to field study approvals (i.e., approving body and any reference numbers):

CITES permit 2012MC/7725

CITES permit 2012MC/7929.

The following information was supplied regarding data availability:

Raw data is available in the Supplemental Information.

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
