# Peer review of "Physiological response of the cold-water coral Desmophyllum dianthus to thermal stress and ocean acidification"

_PeerJ, doi:10.7717/peerj.1606_

## Round 0.1 · original submission · Major Revisions

· Academic Editor

Major Revisions

Both reviewers raise important issues with your introduction, experimental design and discussion. Please incorporate their suggestions, which will improve the quality of your contribution.

Reviewer 1 ·

Basic reporting

No Comments

Experimental design

No Comments

Validity of the findings

No Comments

Additional comments

This is a concise manuscript that suits well within the scope of PeerJ, addressing the very important issue of how future climatic conditions may impact metabolic responses of cold-seawater corals. There are important issues related to the writing structure that requires clean up as few sections are rather well-written but most need a lot of attention. I recommend publication of this paper as long as the issues listed below are addressed and the MS is being re-reviewed prior to publication

Specific comments

Abstract: this section is missing two important parts. First, a succinct explanation of the experimental conditions and second, what is the main conclusion? Why are these results relevant in a broader context? Of course without making unsupported claims.

Results: I suggest including a table whit all carbonate chemistry parameters, measured and calculated. This is key supporting information, mainly for long-term experiments.

Discussion: Unfortunately, this paper suffers from a not well-structured discussion section, which frequently provides and highlights as summaries of other results more than building up from the actual study. This section also needs a good clean up and attention on the structure of sentences. I have gave specific comment for some but not all of the writing problems.

I
L7 and L278: I suggest changing “synergic effect” by “synergistic effect”
L9: I understand what you mean when saying “acclimated to elevated CO2” but it can be confusing as you are actually referring to responses after a long-term exposure rather than a short “acclimation” period. I suggest rewording this part of the sentence.
L9-L10: This sentence structure is incorrect. You seem to be saying.… shows/exhibits no changes in calcification and so on?
L12-L13: Reword this sentence keeping a consistent structure, i.e elevated
seawater temperature (15ºC) significantly reduced calcification and combined elevated temperature and CO2 also affected respiration rates of D. dianthus.
L15-L18: This sentence needs careful rewording
L47: I suggest adding the word “however” to connect with the previous sentence. As it is, doesn’t flow. …Less is currently know, however, on the effects….
L47-53: Such a long sentence with so much information. It needs to be split.
L55: I suggest adding, “For instance, elevated temperature actually….
L58-L62: This sentence needs rewriting
L69: What is “this”? It was way before you mentioned calcification rates. I suggest adding…whether such response can be sustained indefinitely…
L69-L72: This sentence needs careful rewording and more precise information based on the references cited. First, it sounds weird to state that “the acclimation response” may affect metabolism by…. Again, please be very careful in the way you use or misuse the word acclimation.
L82: Use temperature and CO2 rather than temperature and pH, as you didn’t manipulate this last one. Keep consistency in the approach of your experimental conditions through the whole MS.
L97-L101: Rearrange the sentence to separate the ambient condition information from the predicted future conditions. For instance: …..conditions that replicated ambient temperature and CO2 (12°C - 380 ppm) levels and predicted future….
L170-L184: Although it seems to be pretty obvious, I think it is important to include post-hoc tests when significant individual effects are found.
L174-L175: Include the one-way ANOVA information
L187-L191: I suggest the first sentence of this section only mention findings in the present study but most importantly the main and novel findings: synergistic effect and long-term exposure (mentioned in your third paragraph of discussion). I think you also need to keep the same format every time you discuss physiological responses identified in your study.
L202-L204: This sentence needs rewriting. It seems to be missing words to complete the idea.
L213: for a time? Or you mean for a short time?
L215-L219: Reword this sentence so your writing reads well
L261: a missing word….to fulfill increased energy demand AND to maintain…
L266: even if CWC (cold-seawater corals seem instead of seems)… here again you first should promote your findings and not other studies.
L272-L275: Reword this sentence so the read flows

·

Basic reporting

Notes on the Abstract/Introduction:
Some information is missing from the Abstract and Introduction. In the Abstract, you should clarify that you measured instantaneous calcification rate. You also focus a good portion of your abstract space on the O:N ratio, which should be explained in the context of biology in your introduction, and supported with relevant prior studies using the same ratio.
You should cite sources for the natural temperature and pCO2 conditions of your experiment (12ºC and 390ppm), or perhaps provide data on these parameters from your study site if possible.
Why did you use the IPCC IS92a emission scenario (1992)? Why not use the fourth assessment (AR4) published in 2007 (since your experiment was conducted in 2010, and the fifth was not yet available)? Make your reasoning clear in the manuscript and provide a citation.
Most importantly, incomplete raw data was provided. The pCO2, pH, temperature, and salinity data should be attached. The raw data that was provided was already normalized to the control measurements and to the coral skeletal surface area. All of these measurements should be provided prior to any manipulation.
You are missing an acknowledgements section, although I am unsure if this is submitted separately. Also, you should mention permits for coral collection, if required. If not required, see PeerJ policies, Animal Research, #2 “For research conducted on non-regulated animals, a statement should be made as to why ethical approval was not required.”

Experimental design

In general, not enough details on the methods were provided for the experiment to be reproduced, and it is not clear whether the experiment was conducted to a high technical standard. Based on the information given, I do not think the research conforms to prevailing ocean acidification study standards. Below I have included the areas where more information is needed.
When describing your aquarium system, details are essential. You should indicate how long your corals were in tanks prior to the start of your experiment. I.e., how long were they in Monaco, and how long did they acclimate to the new conditions at Heriot-Watt University (prior to being subjected to one of the four treatments)? You need to provide details on the type of lighting you maintained for your corals.
The description of your experimental setup can be clearer. A figure would be very useful here. You should indicate how you assigned subjects to treatments, e.g. randomly. You had 12 colonies total of D. dianthus. In line 101, you say there are three replicate systems of 80L tanks for each treatment, holding three corals each. This implies that you have 36 corals, so you should add language to make it clearer that you had 12 (e.g. one coral per tank). You should also indicate whether the system was open or closed, and what your water source was. Otherwise, it is not possible to understand what is being replicated. In line 103, you say that air and CO2 were bubbled directly into the tanks, but also that gas mixing was accomplished using mixing flasks. A thorough description of the gas-generation system is necessary. Did you mix pure CO2 and room air? Did you use pure CO2 and CO2-free air?
Most importantly, you only provide average pH values and standard deviations (or standard errors? This is not clear) for each of your treatments. In order for this to be an ocean acidification study, you need to characterize the carbonate chemistry of your system. You need to calculate how the carbonate ion concentration is different between your treatments using data for temperature, salinity and two of the following: pH, total alkalinity, total dissolved inorganic carbon, and/or the partial pressure of carbon dioxide in air in equilibrium with a seawater sample (see the Guide to Best Practices for Ocean Acidification Research and Data Reporting, and also Table 1 in Cornwall and Hurd 2015). In addition, your pH values of your controls and experimental treatments overlap. You need to provide data on how the carbonate chemistry changed over time to make a convincing argument that your treatments are distinct and relatively stable. When measuring pH, total alkalinity, etc., you also need to use certified reference materials throughout your experiment to know how precise and accurate your measurements are, and to ensure that all of your equipment is functioning properly. For pH, you need to use TRIS buffers to calibrate your electrode, not NBS.
Some aspects of your response measurements would benefit from additional details. Under “Physiological measurements”, you say four sets of incubations were performed. Does one coral from each treatment make up a “set”? You should make this explicit or just remove this language. For the physiology measurements, you took each of your 12 corals and incubated them individually in 12 incubation chambers. Then you measured instantaneous calcification, respiration, and excretion for each of these corals in each chamber. How did you ensure that your taking measurements for one parameter did not affect the results for another? This should be made clear in the manuscript.
In Line 126 you say that 120 mL seawater samples were drawn before and after incubation. How can you withdraw 240 mL from a chamber of 200 mL volume? You also state that you draw an additional 20 mL before and after incubation from each chamber in Line 150.
The equation starting on line 146 is mg C lost by coral respiration = mg O2 consumed • 0.375RQ (Anthony and Fabricius 2000). You should cite the original reference (Muscatine et al. 1981) and explain any alterations to the equation.
You should briefly describe the spectrofluorometric method you used, since there are two protocols described in Holmes et al. 1999.
You should include the version of R that you used.

Validity of the findings

In order for the differences you observed to be representative of how this species might respond to future ocean acidification, you needed to characterize the carbonate chemistry of your experiment. Without doing this, your interpretation is limited to how this coral species may respond to changes in pH. Also, you state in the discussion section that this species of coral is naturally found at pH levels lower than your elevated pCO2 treatments. This makes it unclear why you did not choose a more extreme pH level. You should clearly state the range of pH levels the corals you used in your experiment are exposed to naturally.
You only describe the O:N ratio at the very end of the discussion, yet it seems to be an important part of your paper based on your abstract. You need to disclose how you calculated the ratio in your materials and methods section, run statistical analyses to test for significant differences between treatments, make a corresponding figure of the results, and then explain whether they are significant in the discussion.
The Discussion section is moderately disjointed with some grammatical errors. Perhaps shortening the paragraphs would improve the flow. In addition, it is somewhat ambiguous what is new about this study compared to previous work. Clearly stating what the novel contributions are to the literature would be beneficial. Finally, at the end of your paper, connect your results back to the big picture. Why should the reader care about reduced calcification and metabolism in D. diathus?

Additional comments

Your results indicating the less efficient protein-dominated energy catabolism in corals subjected to stress are very interesting, and would be a good contribution. I think measuring proxies for coral metabolism are very important in experiments such as this one. However, the experimental design could use some revising.
Based on the above comments, I suggest a thorough review of the available literature on ocean acidification experiments. Particularly, the Guide to Best Practices for Ocean CO2 Measurements (Dickson et al. 2007) and the Guide to Best Practices for Ocean Acidification Research and Data Reporting (Riebesell et al. 2010) are good places to start. Although your experiment is already complete, perhaps you have enough data to properly describe the carbonate chemistry of your treatments. I am unsure based on the raw data you provided.

Notes on Format
-Line 62: change “seems” to “seem”
-Line 67: change “D. diantus” to “D. dianthus”
-Line 77: “focused” is misspelled
-Line 156: “Advanced Geometry” should be capitalized since it is the name of a formal technique.
-Line 171: Change “chambers” to “chamber”, since you had one control beaker.
-Line 192: define what you mean by mid-term
-Line 203: switch the order of the words “slow” and “very”
-Line 227: Change “others” to “other”
-Line 234: Remove the word “be”
-Line 266: Change “seems” to seem, and change “capable” to “able”
-Line 267: Change “effect” to “effects”
-The sentence starting on line 269 with “Even a slight..” and ending on line 272 does not make sense.
-Line 536: Add the word “levels” behind “pCO2”
-Line 537: “as the result of coral nubbins incubation” does not make sense.

---

## Round 0.2 · accepted · Accept

· Academic Editor

Accept

Congratulation on your contribution. I look forward to seeing your paper published in PeerJ soon.